# Effect of exercise preconditioning on myocardial content of Sphingosine1-phosphate and its mechanism in rats after exhaustive exercise

Xinnuan Wei[1], Weiyuan Yang[1]*, Junxiang Zhou[1], Luoyuan Cao[2], Wenxing Jiang[1]

**1** Ningde Clinical Medical College of Fujian Medical University, Ningde, Fujian, People's Republic of China, **2** Department of Rehabilitation Medicine, Ningde Municipal Hospital of Ningde Normal University, Ningde, Fujian, People's Republic of China

* yangwy201509@163.com

## Abstract

### Objective

This study aimed to investigate the effects of exercise preconditioning on rat myocardial Sphingosine1-phosphate(S1P) content and its potential mechanisms of heart protection.

### Methods

A rat model of exercise preconditioning followed by exhaustive exercise was established. Rats were randomized to four groups: control (C), exercise preconditioning (EP), EP plus the S1PR1-selective antagonist W146 (EP+W146), and EP plus the MEK1/2 inhibitor PD98059 (EP+PD98059). Following a final exhaustive swim, comparisons across groups revealed that EP attenuated myocardial injury and apoptosis, an effect which was abolished by both W146 and PD98059.

### Results

1. Exercise preconditioning (EP) significantly attenuated exhaustive exercise-induced myocardial injury and apoptosis ($P < 0.001$); 2. EP significantly elevated myocardial S1P levels ($P = 0.002$), and S1PR1-selective antagonist (W146) abolished this cardioprotective effect ($P = 0.016$ for apoptosis); 3. Most importantly, MAPK pathway inhibition (PD98059) abrogated the protective effect of EP, as evidenced by significantly increased apoptosis ($P = 0.002$), despite unaltered S1P levels.

### Conclusion

In summary, beyond confirming S1P elevation with exercise preconditioning, our findings propose the S1P→MAPK signaling axis as a novel mechanistic pathway warranting future validation.

**Data availability statement:** All data generated or analysed during this study are available in the Figshare repository: [Dataset for "Effect of Exercise Preconditioning on Myocardial Content of Sphingosine1-phosphate and its Mechanism in Rats after Exhaustive Exercise "] (DOI: [https://doi.org/10.6084/m9.figshare.30473615]).

**Funding:** This work was supported by the General Program of Fujian Provincial Natural Science Foundation, China (Grant No. 2024J01941) to Xinnuan Wei.

**Competing interests:** The authors have declared that no competing interests exist.

## Introduction

Exercise preconditioning, characterized by repeated brief intermittent high-intensity exercise, enhances the heart's tolerance to prolonged ischemic hypoxia and is considered effective in reducing myocardial ischemic injury. Mechanisms include improvements in calcium handling and upregulation of endogenous antioxidant enzymes [1,2]. Sphingosine1-phosphate (S1P) has been shown to protect cardiomyocytes during ischemic preconditioning and postischemic conditioning [3], potentially through mechanisms involving upregulation of endogenous antioxidant enzymes and improved calcium handling [4]. Previous reports indicate that endurance exercise may upregulates sphingosine kinase 1 (SPHK1) activity, enhancing S1P synthesis. S1P binding to receptors (e.g., S1PR1/S1PR2) activates downstream MAPKs (ERK1/2 and p38), which suppress pro-apoptotic proteins, promote mitophagy, and bolster antioxidant defenses [5]. Therefore, this experiment aims to investigate whether exercise preconditioning exerts myocardial protective effects via activation of the MAPK signaling pathway by increasing S1P content, using an animal model of exercise preconditioning and exhaustive exercise.

## Materials and methods

### 1.1. Primary reagents and kits

S1P, the S1PR1-selective antagonist W146, and the MEK1/2 inhibitor PD98059 were obtained from GlpBio Corporation (USA). The terminal deoxynucleotidyl transferase dUTP nick end labeling (TUNEL) kit and the Annexin V-FITC/PI apoptosis detection kit were purchased from Roche (Germany).The S1P ELISA kit (Catalog No: UCE-G013Ge) was acquired from Wuhan Yunclone (China).

### 1.2. Experimental animals and ethics statement

**1.2.1. Experimental animals.** Twenty-two healthy male Sprague-Dawley (SD) rats, weighing between 200 and 220 g, were housed under routine conditions with 5 animals per cage. They were provided with ad libitum access to water and standard rodent diet, maintained at a room temperature of 20−22°C with a relative humidity of 45%−50% and a 12-hour light-dark cycle. The SD rats and their feed were supplied by Wu's Experimental Animal Center, Fujian Province, under license SCXK (Hong's) 2018−0006. All procedures involving experimental animals were conducted in accordance with the Guiding Opinion on the Ethical Treatment of Experimental Animals issued by the Ministry of Science and Technology of the People's Republic of China [6].

**1.2.2. Methods of sacrifice.** At the conclusion of the experiment, all rats were humanely euthanized using carbon dioxide ($CO_2$) inhalation, in accordance with the American Veterinary Medical Association (AVMA) Guidelines for the Euthanasia of Animals (2020). The $CO_2$ euthanasia process was conducted as follows: (1) Animals were placed in a well-ventilated euthanasia chamber. (2)$CO_2$ was gradually introduced into the chamber at a flow rate of 3–70% of the chamber volume per minute to avoid rapid asphyxiation. (3)Euthanasia was confirmed by monitoring

physiological signs (cessation of breathing, absence of heartbeat, and corneal reflex) for a minimum of 5 minutes post-loss of consciousness. (4)Secondary verification of death was performed by cervical dislocation.This method was selected for its rapid onset of unconsciousness, minimizing distress, and its compliance with institutional ethical guidelines. The euthanasia procedure was performed by trained personnel with appropriate certification in animal handling and euthanasia techniques.

**1.2.3. Efforts to alleviate suffering.** To ensure the welfare of animals and minimize suffering throughout the study, the following measures were implemented:(1)Experimental Design & 3R Principles: Reduction: The sample size was optimized through statistical power calculations to minimize animal numbers while maintaining scientific validity. Refinement: Exercise protocols were designed to gradually acclimate animals to swimming (1-week acclimation period with 15-minute daily weightless swimming sessions) to reduce stress associated with forced exercise.Rationale for Exercise Model: The swimming protocol was selected as a non-invasive, whole-body exercise model with established scientific validity for studying physiological adaptations. (2)During Experimentation:Progressive exercise intensity: The tail-loaded swimming exercise (3% body weight) was introduced only after animals demonstrated physical adaptation during the acclimation phase.Real-Time monitoring: Trained observers continuously monitored animals during the exhaustive swimming session to identify signs of distress (e.g., dysregulated swimming actions, prolonged submersion). The "exhaustion endpoint" was defined by the inability to maintain buoyancy for >5 seconds, at which point the animal was immediately removed from the water to prevent drowning-related suffering.Environmental enrichment: Rats were housed in groups in environmentally enriched cages (including bedding material, chew toys, and hiding spaces) to promote natural behaviors and reduce stress.(3) Post-Exercise & Post-Sacrifice Care: Immediately after the exhaustive swimming session, animals were gently dried with towels and placed in a warm recovery area (maintained at 28–30°C) to prevent hypothermia.Animals were monitored for signs of pain or distress post-exercise; analgesics (e.g., buprenorphine) were administered as needed under veterinary supervision.Euthanasia was performed under deep anesthesia (using pentobarbital) when necessary to ensure painless transition.

**1.2.4. Ethics statement.** All animal experiments were approved by the Institutional Animal Care and Use Committee (IACUC) of NingDe Municipal Hospital of Ningde Normal University (Approval No.: IACUC-20200310) on March 21, 2020. All procedures were performed in strict accordance with the National Standard of China (GB/T 35892-2018) and followed the ARRIVE guidelines 2.0 as well as the U.S. National Institutes of Health Guide for the Care and Use of Laboratory Animals.

## 1.3. Experimental design, grouping, and animal models

The sample size of n = 5–6 per group was determined for this exploratory study. As this research was designed to investigate novel mechanistic insights into the S1P/MAPK pathway, a formal a priori power calculation was not feasible. The sample size was selected based on common practices in preliminary physiological research and practical feasibility, with the goal of identifying robust effect trends to inform future hypothesis-driven investigations. The rats were acclimatized for 1 week and subjected to daily 15-minute sessions of weightless swimming exercise. After this adaptation period, rats were randomized into intervention and control groups using a computer-generated randomization list. The random sequence was generated using computer-generated block randomization (block size 4–6) via SAS 9.4, stratified by body weight and baseline measurements. Allocation concealment was ensured using sequentially numbered, sealed, opaque envelopes prepared by a statistician not involved in the experiment. Envelopes were opened only after the rat was enrolled and baseline data recorded. An independent researcher, unrelated to animal housing or treatment, performed the allocation. Randomization occurred immediately before intervention initiation. And 22 rats were randomly divided into the following groups: normal control (group C, n = 5), exercise-preconditioning (EP group, n = 5), EP with the S1PR1-selective antagonist W146 (EP + W146, n = 6), and EP with the MEK1/2 inhibitor PD98059 (EP + PD98059, n = 6).The group C was maintained under routine conditions for 3 weeks. The remaining groups underwent intermittent swimming with a tail-loaded

weight equivalent to 3% of their body mass to establish long-term exercise-preconditioned animal models, following the method described by Margonato et al [7]. This involved daily sessions of non-weight-bearing swimming for 15 minutes, followed by tail-loaded swimming in a pool measuring 40 cm deep and 50 cm in diameter at a temperature of $37 \pm 2°C$. The exercise duration gradually increased from 30 minutes per day in week 1, 1 hour per day in week2, and 2 hours per day in week 3, conducted 6 days per week. The swimming intensity (e.g., swim speed or resistance) does not progressively increased along with the duration. The EP+W146 group received an intraperitoneal injection of W146 (20 mg/kg) 10 min prior to the commencement of each exercise session. The EP+PD98059 group also received PD98059 (100 mg/kg) via identical timing and route (i.p., 10 min before exercise). All inhibitors were given before every exercise session, with doses adjusted according to body weight. All reagents were prepared in normal saline as the vehicle.The preconditioning regimen for each treatment group was identical to the exercise protocol. Three weeks after the experiment commenced, rats in the groupC, EP group, EP+W146 group, and EP+PD98059 group underwent a single bout of exhaustive swimming exercise with a tail-loaded weight equivalent to 3% of their body mass. Exhaustion was determined by evident swimming fatigue, extremely slow paddling speed, continuous sinking, and an inability to persist in swimming.

## 1.4. Euthanasia and tissue collection

Following euthanasia, which was performed thirty minutes post-modeling as detailed in section 1.2.2, tissue and blood samples were collected for subsequent analysis. Immediately thereafter, blood was drawn via cardiac puncture, and the hearts were rapidly excised, weighed, and rinsed with sterile saline. A portion of the myocardial tissue was snap-frozen for biochemical assays, while another portion was fixed for histological examination.

## 1.5. Biochemical and histological analyses

### 1.5.1. Quantification of myocardial S1P.
Myocardial S1P levels were quantified using a competitive ELISA approach. Tissue samples (~100 mg) were homogenized in ice-cold phosphate-buffered saline (1:10, w/v) using a mechanical homogenizer. The homogenate was centrifuged at $3,000 \times g$ for 10 min at 4°C, and the supernatant was collected for analysis. S1P concentrations were determined using a commercial ELISA kit according to the manufacturer's protocol, with absorbance measurements performed on a BioTek microplate reader (Bolton Company, USA). The assay detection range was 12.35–1000 ng/mL, with a sensitivity of ≤4.73 ng/mL. Total protein concentration in the supernatant was measured using a bicinchoninic acid (BCA) assay kit (Thermo Fisher Scientific, USA). Final S1P levels were normalized to total protein content and expressed as ng/mg protein.

### 1.5.2. Histopathological analysis.
Myocardial tissues were fixed in 10% neutral buffered formalin for 24 hours, routinely processed through a graded ethanol series and xylene, and embedded in paraffin. Sections (4–5 µm thick) were deparaffinized, rehydrated, and stained with hematoxylin and eosin (H&E) following standard procedures. Specifically, nuclei were stained with hematoxylin, differentiated in acid alcohol, and blued in tap water. Cytoplasmic counterstaining was performed with eosin. Finally, sections were dehydrated, cleared in xylene, and mounted. All sections were examined under a light microscope (NIKON ECLIPSE E100, Japan) at 400×magnification by two experienced pathologists blinded to the experimental groups to assess histological features such as cellular morphology and tissue architecture.

### 1.5.3. TUNEL assay.
Apoptosis in myocardial tissues was detected using the Terminal deoxynucleotidyl transferase dUTP nick end labeling (TUNEL) assay, performed according to the manufacturer's protocol. Nuclei of apoptotic cardiomyocytes were stained brown, while non-apoptotic nuclei appeared blue. For each group, three randomly selected sections were analyzed. From each section, three non-overlapping fields were captured at 200×magnification. The apoptotic index was calculated as the percentage of TUNEL-positive nuclei relative to the total number of nuclei in the field. To ensure specificity, positive controls (tissue sections treated with DNase) and negative controls (sections incubated without terminal deoxynucleotidyl transferase enzyme) were included in each assay batch. Automated image analysis

with predefined thresholds for nuclear size and staining intensity was used for quantification, and results were verified independently by two blinded observers.

**1.5.4. Western blot for p-MEK.** The expression of p-MEK was evaluated by Western blot analysis. Protein samples were separated by SDS-PAGE, transferred onto PVDF membranes, and probed with specific primary and secondary antibodies. p-MEK levels were normalized to the loading control (e.g., GAPDH) and quantified using ImageJ software (National Institutes of Health, Bethesda, MD, USA).

## 1.6. Blind analysis protocol

To minimize bias in data acquisition and analysis, an assessor-blinded study design was implemented throughout the experimental procedures. Personnel responsible for conducting biochemical assays (ELISA, Western blot), performing histopathological evaluations, and scoring TUNEL assays were unaware of the group assignments until all quantitative analyses were completed.

## 1.7. Statistical analysis

The concentration of S1P was designated as the single primary endpoint. All statistical analyses were performed using SPSS (version 23.0, IBM, Armonk, NY, USA) and GraphPad Prism (version 10.0, GraphPad Software, San Diego, CA, USA). The normality of all continuous variables was confirmed using the Shapiro-Wilk test. Consequently, data are presented as mean ± standard deviation (SD). For the comparison of the primary endpoint (S1P) and other continuous variables between two groups, the independent two-tailed Student's t-test was employed. For comparisons across more than two groups, one-way analysis of variance (ANOVA) was applied, followed by Tukey's post-hoc test for multiple comparisons. The assumption of homogeneity of variances for ANOVA was verified using Levene's test. For all key comparisons, effect sizes (Cohen's $d$ for t-tests and $\eta^2$ for ANOVA) along with their 95% confidence intervals (95% CI) are reported. All tests were two-sided, with statistical significance set at $P < 0.05$.

## Results

### 2.1. Exercise preconditioning attenuates myocardial injury induced by exhaustive exercise in rats

As shown in Fig 1, exercise preconditioning (EP) provided significant protection against exhaustive exercise-induced myocardial injury. Cardiac hypertrophy was not a contributing factor, as indicated by the comparable heart weights between group C and the EP group ($P = 0.388$; Fig 1A and Table 1). Instead, the cardioprotective effects of EP were evidenced at the tissue and cellular levels. Histological analysis of H&E-stained sections demonstrated markedly improved myocardial preservation in the EP group, characterized by reduced cardiomyocyte rupture and interstitial changes (Fig 1B). Correspondingly, TUNEL assay revealed that EP significantly suppressed exhaustive exercise-induced cardiomyocyte apoptosis ($P < 0.001$; Fig 1C and 1D and Table 2).

### 2.2. Exercise preconditioning mediates cardioprotection during exhaustive exercise in rats via enhanced sphingosine-1-phosphate (S1P) generation

**2.2.1. Exercise preconditioning upregulates myocardial S1P content in rats subjected to exhaustive exercise.** Following exhaustive exercise, the myocardial S1P content in the EP group was significantly higher than in group C ($P = 0.002$; Fig 2A and Table 3), with a large effect size (Cohen's $d = 2.79$, 95% CI [−4.25, −1.34]).

**2.2.2. S1P attenuates exhaustive exercise-induced eyocardial injury in rats.** Administration of the S1PR1-selective antagonist W146 significantly abrogated the cardioprotective effects of exercise preconditioning. The heart weight of rats in the EP + W146 group was significantly higher than in the EP group ($P = 0.008$; Fig 2B and Table 4). Consistent with this, H&E-stained sections revealed more prominent cardiomyocyte rupture and interstitial edema in the

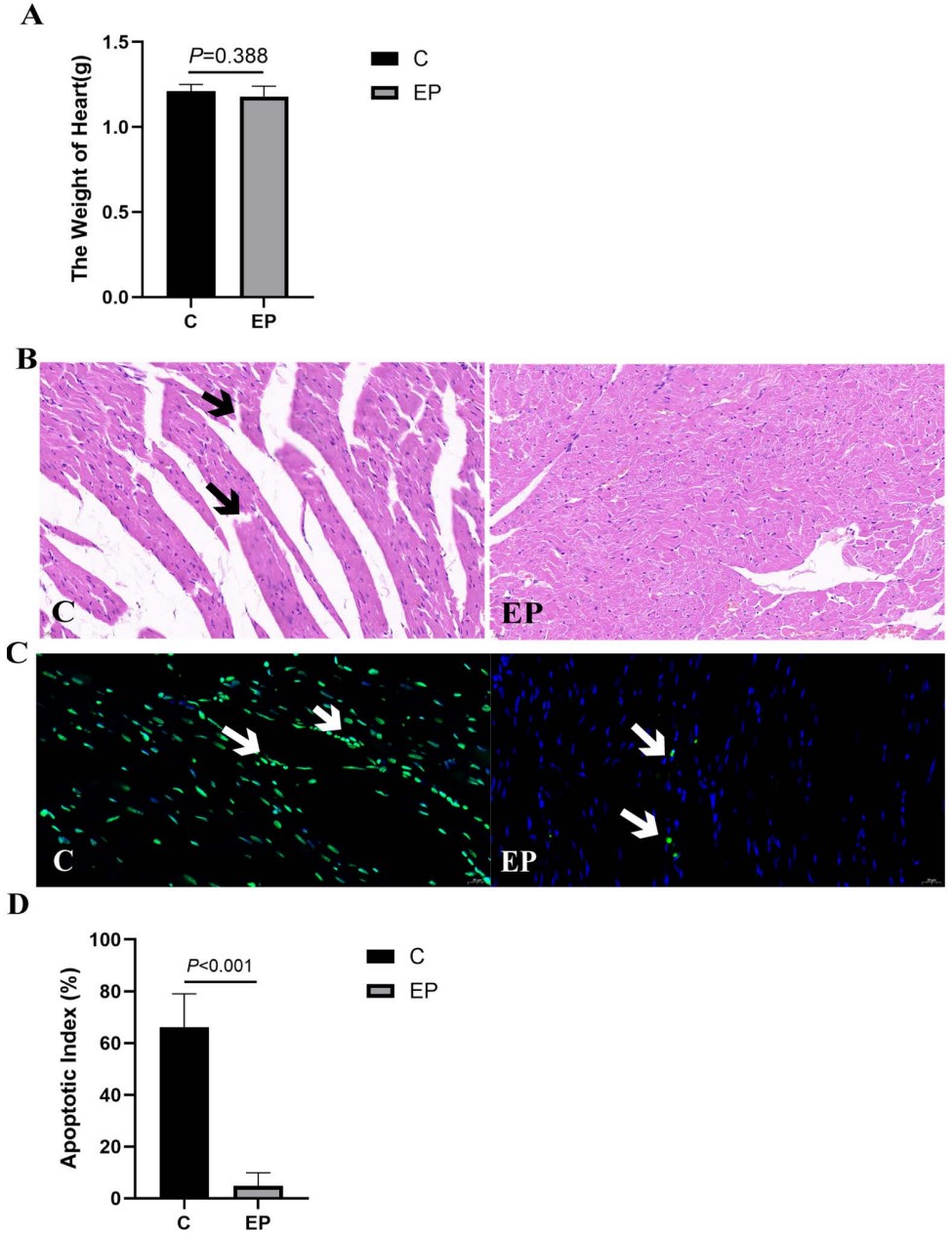

**Fig 1. Exercise preconditioning attenuates myocardial injury induced by exhaustive exercise in rats. (A)** Heart weights of rats. Data are presented as mean±SD (n=5 per group). An independent samples Student's t-test was used for comparison between group C and the EP group (*P*=0.388, Cohen's *d*=0.58, 95% CI [–0.88, 2.04]). **(B)** Representative images of H&E-stained rat heart tissue sections observed at 400×magnification. The EP group exhibited reduced cardiomyocyte rupture and interstitial alterations compared with group **C.** Scale bar: 25 µm. **(C & D)** Cardiomyocyte apoptosis induced by exhaustive exercise was evaluated by TUNEL staining at 400×magnification. Data are expressed as mean±SD (n=5 per group) (n=5 per group), and between-group comparisons were performed using the independent samples Student's t-test. The EP group showed significantly lower apotosis index than group C (*P*<0.001, Cohen's d=6.33, [2.75,9.90]). Scale bar: 25 µm.

**Table 1. Comparison of heart weight between group C and the EP group.**

| Variable & (Group) | n | Mean±SD | Statistical Model | P Value | Effect Size | 95% CI |
|---|---|---|---|---|---|---|
| The Weight of Heart (g) | | | t-test | 0.388 | Cohen's d=0.58 | [−0.88 2.04] |
| Group C | 5 | 1.21±0.04 | | | | |
| EP Group | 5 | 1.18±0.06 | | | | |

**Table 2. Comparison of the myocardial apoptotic index between group C and the EP group.**

| Variable & (Group) | n | Mean±SD | Statistical Model | P Value | Effect Size | 95% CI |
|---|---|---|---|---|---|---|
| Apoptotic Index(%) | | | t-test | <0.001 | Cohen's d=6.33 | [2.75,9.90] |
| Group C | 5 | 66.20±12.77 | | | | |
| EP Group | 5 | 4.80±5.03 | | | | |

EP+W146 group compared to the EP group (Fig 2C). Furthermore, TUNEL assay demonstrated a significant increase in exhaustive exercise-induced cardiomyocyte apoptosis in the EP+W146 group ($P=0.016$; Fig 2D, 2E and Table 5).

### 2.3. S1P protects against exhaustive exercise-induced myocardial injury in rats via the MAPK pathway

**2.3.1. MEK1/2 inhibition abrogates S1P-mediated exercise preconditioning.** Administration of the MEK1/2 inhibitor PD98059 revealed a critical role for the MAPK pathway. A one-way ANOVA identified a significant main effect of group on p-MEK expression ($F(3,8) = 113.74$, $P<0.0001$, $\eta^2=0.977$), as shown in Fig 3A. Post-hoc testing confirmed that the exhaustive exercise-induced increase in p-MEK observed in the EP group was significantly abrogated in the EP+PD98059 group ($P<0.05$). In contrast, myocardial S1P content after exhaustive exercise did not differ significantly between these two groups ($P=0.73$; Fig 3B and Table 6).

**2.3.2. Role of MEK1/2 signaling in exercise preconditioning-induced cardioprotection.** Inhibition of MEK1/2 with PD98059 had no significant effect on heart weight compared to the EP group alone ($P=0.849$; Fig 3C, Table 7). However, it exacerbated exhaustive exercise-induced myocardial injury, as evidenced by more severe cardiomyocyte rupture and interstitial changes (Fig 3D) and a significant increase in cardiomyocyte apoptosis ($P=0.002$; Fig 3E, 3F, Table 8).

## Discussion

Exercise preconditioning, characterized by repeated transient intermittent high-intensity exercise, enhances the heart's resilience against prolonged ischemic hypoxia and is considered effective in mitigating myocardial ischemic damage [8]. Several studies have demonstrated that exercise preconditioning can alleviate myocardial damage in rats subjected to exhaustive exercise, evident as reduced myocardial hypertrophy, myocardial rupture, and myocardial interstitial hyperplasia [9,10]. In this experiment, HE staining revealed significantly less myocardial rupture and interstitial hyperplasia in rats subjected to exercise preconditioning compared to those without preconditioning. However, there was no significant difference in heart weight between the exercise-preconditioned group and the control group, suggesting a potential balance between myocardial apoptosis, hypertrophy, and interstitial hyperplasia in the control group. Additionally, exercise preconditioning may exert myocardial protective effects by reducing cardiomyocyte free radical release [9] and up-regulating LC3 protein via other pathways [11].

Studies have demonstrated the protective role of S1P for cardiomyocytes in ischemic preconditioning and postischemic conditioning [3]. The findings of this experiment indicate that myocardial S1P content in exhausted rats subjected to exercise preconditioning is significantly higher compared to the control group, consistent with Morrell MBG et al.'s findings [12]. The mechanism by which exercise influences S1PR1 protein content involves central actions of IL-6 in the

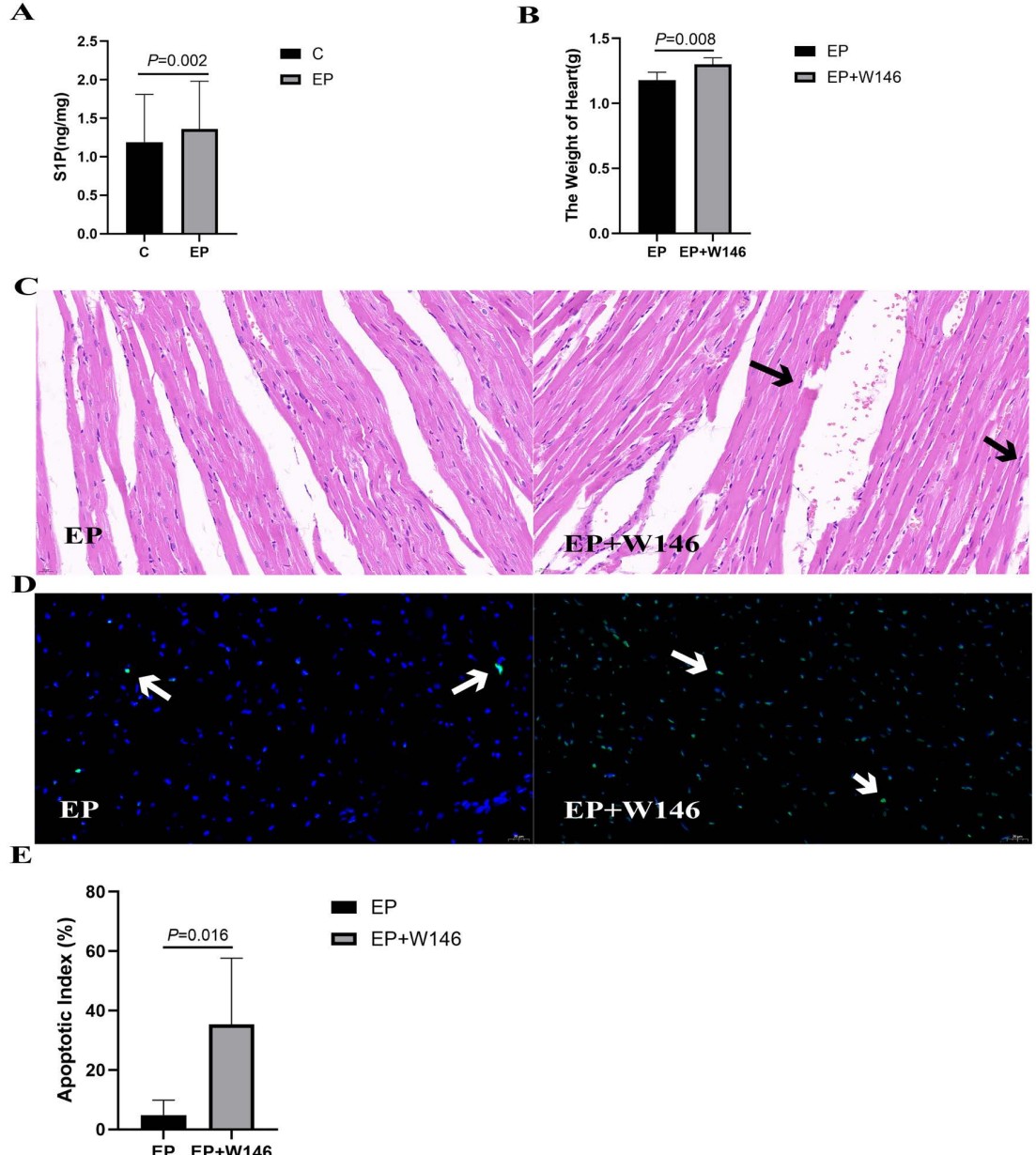

**Fig 2. S1P protects against exhaustive exercise-induced myocardial injury in rats.** (A) Myocardial S1P concentration quantified by ELISA. Data are presented as mean±SD (n=5 per group). An independent samples Student's t-test was used for comparison between group C and the EP group, the EP group showed a significantly higher myocardial S1P concentration than group C($P$=0.002, Cohen's $d$=2.79, 95% CI [1.34, 4.25]).(B) Heart weights of rats. Data are expressed as mean±SD (n=5−6 per group). An independent samples t-test revealed that the EP+W146 group had significantly heavier hearts than the EP group($P$=0.008, Cohen's $d$=2.03, 95% CI [0.04, 0.19]). (C) Representative H&E-stained sections of rat myocardium (400×). The EP+W146 group exhibits increased cardiomyocyte rupture and interstitial changes relative to the EP group. Scale bar: 25 μm. (D & E) Cardiomyocyte apoptosis assessed by TUNEL staining. Data are shown as mean±SD (n=5−6 per group). The EP+W146 group exhibited a significantly higher cardiomyocyte apoptosis index than the EP group, as determined by an independent samples Student's t-test ($P$=0.016, Cohen's $d$=1.80, [−3.24, −0.18]). Scale bar: 25 μm.

**Table 3. Myocardial S1P concentration in group C and EP-group rats after exhaustive exercise.**

| Variable & (Group) | n | Mean±SD | Statistical Model | P Value | Effect Size | 95% CI |
|---|---|---|---|---|---|---|
| S1P(ng/mg) | | | t-test | 0.002 | Cohen's d=2.79 | [−4.25, −1.34] |
| Group C | 5 | 1.19.±0.62 | | | | |
| EP Group | 5 | 1.36±0.62 | | | | |

**Table 4. Comparison of heart weight between the EP and EP+W146 groups.**

| Variable | n | Mean±SD | Statistical Model | P Value | Effect Size | 95% CI |
|---|---|---|---|---|---|---|
| Weight of heart (g) | | | t-test | 0.008 | Cohen's d=2.03 | [−0.19, −0.04] |
| EP Group | 5 | 1.18±0.06 | | | | |
| EP+W146 Group | 6 | 1.30±0.05 | | | | |

**Table 5. Comparison of the myocardial apoptotic index between the EP and EP+W146 groups.**

| Variable & (Group) | n | Mean±SD | Statistical Model | P Value | Effect Size | 95% CI |
|---|---|---|---|---|---|---|
| Apoptotic Index (%) | 5 | | t- test | 0.016 | Cohen's d=1.80 | [−3.24, −0.18] |
| EP Group | | 4.80±5.03 | | | | |
| EP+W146 Group | 6 | 35.30±22.32 | | | | |

hypothalamus [13], increased plasma S1P content through exercise, leading to elevated S1P levels in skeletal muscle [14], and exercise-induced release of vascular endothelial cell-derived S1P [15,16].

The results further revealed that inhibition of S1PR1 with W146 significantly increased the myocardial apoptotic index following exercise preconditioning, suggesting that the S1P/S1PR1 axis is critical for conferring anti-apoptotic protection.. Addition of the S1PR1-selective antagonist significantly increased heart weight, myocyte rupture, and interstitial changes in exhausted rats, collectively indicating that S1P protects myocardium in exhausted rats. Its mechanisms may involve promoting the biological effects of muscle through enhanced expression of S1PRs in skeletal muscle, such as excitation-contraction coupling, activation of satellite cells, and protection via improved mitochondrial functions that enhance skeletal muscle cell adaptability to exercise training [14–17].

Additionally, the results of this study indicate that there was no significant difference in S1P content in exhausted rats between the exercise-preconditioned + the MEK1/2 inhibitor PD98059 groups, suggesting that PD98059 do not markedly reduce S1P levels in this context. Western blot analysis showed significantly weaker p-MEK expression in the EP+PD98059 group compared to the exercise-preconditioned group. HE staining indicated increased myocyte rupture and interstitial changes in the EP+PD98059, suggesting greater myocardial damage compared to the non-MEK1/2 inhibitor group. And studies show that PD98059 exacerbates myocardial injury in exercise-precondition by blocking survival signals, inhibiting endogenous protectants and aggravating oxidative stress, underscoring the indispensable role of MAPK in adaptive cardioprotection,which can explain why apoptosis increases [18,19]. These findings suggest that S1P likely plays a role in myocardial protection through MAPK signaling pathways independent of affecting S1P content.

A key question in S1P-MAPK biology is whether their relationship constitutes a bidirectional feedback loop or can exhibit a dominant directional flow in specific physiological contexts. Although prior work in endurance adaptation has described a robust S1P-MAPK positive feedback loop [5,16], our findings in an exercise preconditioning model provide evidence for the latter. The demonstration that MAPK inhibition completely abolishes protection while leaving S1P levels unchanged allows us to disentangle this interplay, positioning MAPK activation as an essential downstream

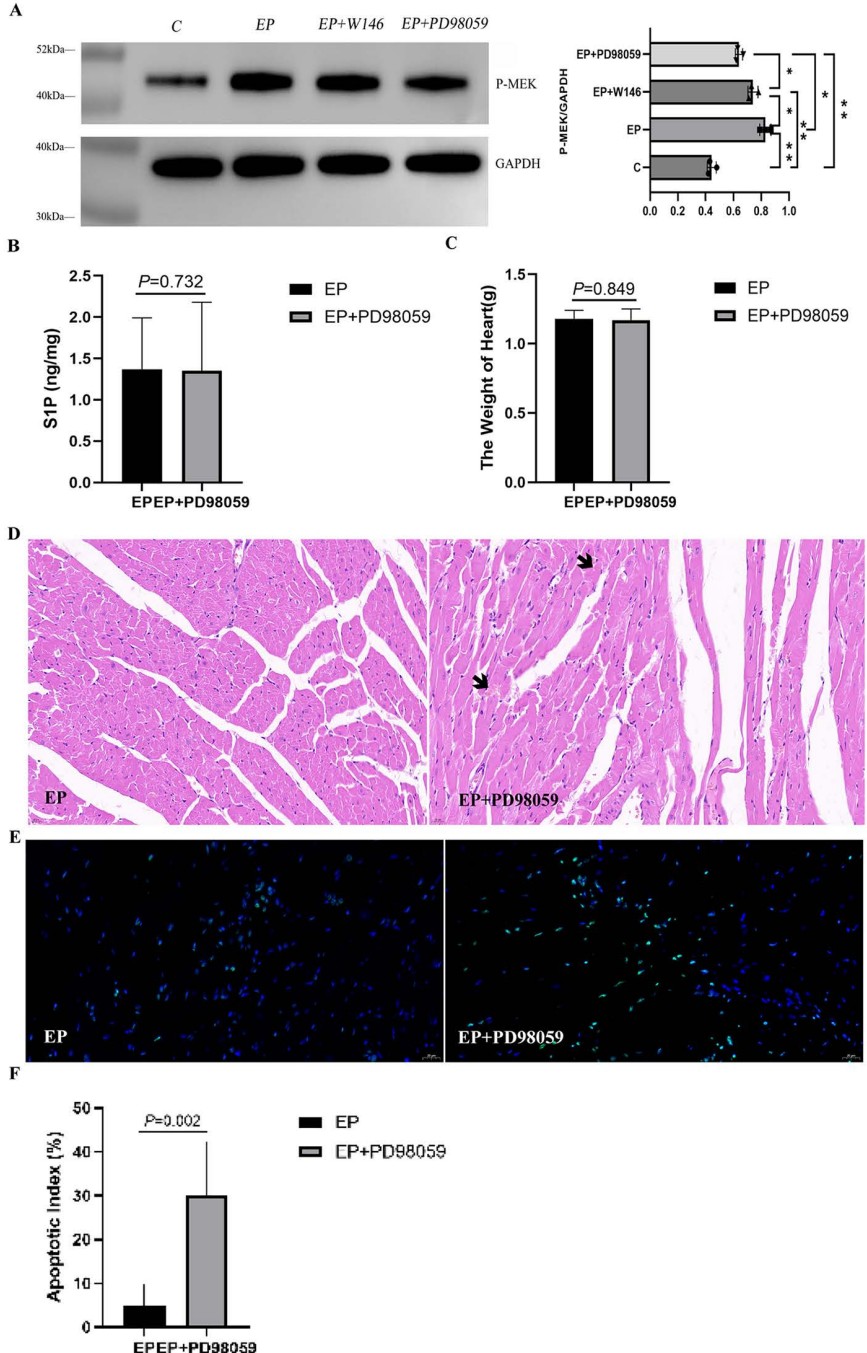

**Fig 3. S1P exerts a protective effect in the rat myocardium against exhaustive exercise via the MAPK signaling pathway.** (A) Expression of p-MEK protein. Representative Western blots and densitometric analysis are shown. Data are presented as mean±SD (n=3 per group). One-way ANOVA revealed a significant main effect of group ($F(3,8) = 113.74$, $P<0.0001$, $\eta^2=0.98$). Post-hoc Tukey's test showed that the p-MEK level in the EP group was significantly higher than that in the EP+PD98059 group (*$P<0.05$), **$P<0.001$ vs. other relevant groups. (B) Myocardial S1P content. S1P was quantified by ELISA. Data are expressed as mean±SD (n=5–6 per group). An independent samples Student's t-test showed no significant difference between the EP and EP+PD98059 groups ($P=0.73$, Cohen's $d=0.04$, 95% CI [−1.42, 1.50]).(C) Heart weights of rats. Data are expressed as mean±SD (n=5–6 per group). An independent samples Student's t-test showed no significant difference between the EP and EP+PD98059 groups ($P=0.849$, Cohen's $d=0.12$, 95% CI [−1.25, 1.49]). (D) Representative H&E-stained sections of rat myocardium (400×). The EP+PD98059 group exhibits increased cardiomyocyte rupture and interstitial changes compared to the EP group. Scale bar: 25 μm. (E & F) Cardiomyocyte apoptosis assessed by TUNEL staining. Data are expressed as mean±SD (n=5–6 per group). An independent samples Student's t-test indicated a significantly higher apoptosis index in the EP+PD98059 group than in the EP group ($P=0.002$, Cohen's $d=2.58$,[−4.43, −0.73]). Scale bar: 25 μm.

**Table 6. Comparison of myocardial S1P concentration between the EP and EP+PD98059 groups.**

| Variable & (Group) | n | Mean±SD | Statistical Model | P Value | Effect Size | 95% CI |
|---|---|---|---|---|---|---|
| S1P(ng/mg) | | | t-test | 0.732 | Cohen's d=0.04 | [−1.42, 1.50] |
| EP Group | 5 | 1.37±0.62 | | | | |
| EP+PD98059 Group | 6 | 1.35±0.83 | | | | |

**Table 7. Comparison of heart weight between the EP and EP+PD98059 groups.**

| Variable | n | Mean±SD | Statistical Model | P Value | Effect Size | 95% CI |
|---|---|---|---|---|---|---|
| Weight of heart (g) | | | t-test | 0.849 | Cohen's d=0.12 | [−1.25, 1.49] |
| EP Group | 5 | 1.18±0.06 | | | | |
| EP+PD98059 Group | 6 | 1.17±0.08 | | | | |

**Table 8. Comparison of the myocardial apoptotic index between the EP and EP+PD98059 groups.**

| Variable & (Group) | n | Median [IQR] | Statistical Model | P Value | Effect Size | 95% CI |
|---|---|---|---|---|---|---|
| Apoptotic Index(%) | | | t- test | 0.002 | Cohen's d=2.58 | [−4.43, −0.73] |
| EP Group | 5 | 4.80±5.03 | | | | |
| EP+PD98059 Group | 6 | 30.01±12.33 | | | | |

event contingent upon S1P signaling. This proposes a refined, S1P→MAPK hierarchical model that complements the existing concept of a feedback loop and offers a more precise mechanistic framework for exercise-induced cardioprotection.

Previous studies have reported that activation of the S1P/S1PR1/STAT3 signaling pathway may promote inflammatory damage and fibrosis of the cardiac valves [20]; and the activation of S1P receptors plays a crucial role in inhibiting the activation of Caspase-3, a key executor of apoptosis, by stimulating the AKT pathway [21]. And the studies suggests that eNOS-knockout mice lose S1P-mediated cardioprotection [22]; endothelial S1PR1 regulates pressure overload-induced cardiac remodelling through AKT-eNOS pathway [23]. Whether exercise preconditioning influences S1P synthesis through these pathways awaits further experimental validation.

In summary, while our findings confirm established roles of exercise preconditioning and S1P, they propose a novel hierarchical relationship by suggesting that the MAPK pathway is a necessary downstream component of S1P-mediated protection. This proposed S1P→MAPK axis represents a key mechanistic insight from our work, and future work is warranted to fully validate this signaling hierarchy.

## Supporting information

**S1 File. WB-raw-images.** Contains the complete set of original, uncropped, and unprocessed Western blot images from which the data presented in this study were derived. The lanes containing the molecular weight markers are present. The corresponding molecular weight calibration reference for these markers is provided in S2 File.
(ZIP)

**S2 File. Molecular weight marker reference.** Provides a reference image showing the precise alignment of the molecular weight markers (in kDa) with their corresponding bands in the original Western blot images contained in S1 File. This allows for independent verification of protein sizes presented in the main figures.
(PDF)

**S3 File. Raw H&E source images.** Contains all original, uncropped microscope images supporting the histology results in Fig 1B, Fig 2C, and Fig 3D, provided as a ZIP archive (S3_File RAW HE.zip). The archive includes the following files, with their specific correspondences detailed below: S3 File C.jpg: The 40x field for the Control group panel in Fig 1B. S3 File EP 1.jpg: The 40x field for the EP group panel in Fig 1B. S3 File EP 2.jpg: The 40x field for the EP group panel in Fig 2C. S3 File EP 3.jpg: The 40x field for the EP group panel in Fig 3D. S3 File W.jpg: The 40x field for the EP+W146 group panel in Fig 2C. S3 File P.jpg: The 40x field for the EP+PD98059 group panel in Fig 3D. S3 File EP Source.jpg: The low-magnification (3x) source image from which the three EP group 40x fields (EP_1, EP_2, EP_3) were cropped. S3 File EP Guide.tif: An annotated guide image. The locations of the cropped 40x fields are outlined and labeled as EP_1 (for Fig 1B), EP_2 (for Fig 2C), and EP_3 (for Fig 3D) within this source image.
(ZIP)

**S4 File. Supplementary representative histological images (H&E staining).** This file contains additional, representative H&E-stained images from the study that were not featured in the main figures but provide further context and demonstrate the consistency of observations within each experimental group. The images cover all four groups: Control (C), EP group (EP), EP+W146 (W), and EP+PD98059(P). Both low-magnification (3x) overviews and high-magnification (40x) detail views are included where available. These supplementary images support the robustness and generalizability of the histological findings presented in the manuscript.
(ZIP)

## Author contributions

**Conceptualization:** Xinnuan Wei.

**Data curation:** Xinnuan Wei, Junxiang Zhou.

**Formal analysis:** Junxiang Zhou.

**Methodology:** Weiyuan Yang.

**Project administration:** Luoyuan Cao.

**Resources:** Luoyuan Cao.

**Software:** Wenxing Jiang.

**Supervision:** Weiyuan Yang.

**Writing – original draft:** Xinnuan Wei.

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
