## [Decision Letter · Decision Letter 0]

16 Jul 2025

Dear Dr. Yang,

Thank you for submitting your manuscript to PLOS ONE. After careful consideration, we feel that it has merit but does not fully meet PLOS ONE’s publication criteria as it currently stands. Therefore, we invite you to submit a revised version of the manuscript that addresses the points raised during the review process.

We look forward to receiving your revised manuscript.

Kind regards,

Amirreza Khalaji

Academic Editor

PLOS ONE

Journal Requirements:

2. To comply with PLOS ONE submissions requirements, in your Methods section, please provide additional information regarding the experiments involving animals and ensure you have included details on (1) methods of sacrifice, and (2) efforts to alleviate suffering.

“This work was supported by the General Program of Fujian Provincial Natural Science Foundation (2020J011350).”

5. We note that your Data Availability Statement is currently as follows: All relevant data are within the manuscript and in Supporting Information files.

7. Your ethics statement should only appear in the Methods section of your manuscript. If your ethics statement is written in any section besides the Methods, please move it to the Methods section and delete it from any other section. Please ensure that your ethics statement is included in your manuscript, as the ethics statement entered into the online submission form will not be published alongside your manuscript.

8. Please include your tables as part of your main manuscript and remove the individual files. Please note that supplementary tables (should remain/ be uploaded) as separate "supporting information" files.

9. PLOS ONE now requires that authors provide the original uncropped and unadjusted images underlying all blot or gel results reported in a submission’s figures or Supporting Information files. This policy and the journal’s other requirements for blot/gel reporting and figure preparation are described in detail at https://journals.plos.org/plosone/s/figures#loc-blot-and-gel-reporting-requirements and https://journals.plos.org/plosone/s/figures#loc-preparing-figures-from-image-files. When you submit your revised manuscript, please ensure that your figures adhere fully to these guidelines and provide the original underlying images for all blot or gel data reported in your submission. See the following link for instructions on providing the original image data: https://journals.plos.org/plosone/s/figures#loc-original-images-for-blots-and-gels.  

Reviewers' comments:

Reviewer's Responses to Questions

**Comments to the Author**

1. Is the manuscript technically sound, and do the data support the conclusions?

Reviewer #1: No

Reviewer #2: Yes

Reviewer #3: Yes

Reviewer #4: Partly

2. Has the statistical analysis been performed appropriately and rigorously?

Reviewer #1: Yes

Reviewer #2: Yes

Reviewer #3: Yes

Reviewer #4: N/A

3. Have the authors made all data underlying the findings in their manuscript fully available?

Reviewer #1: No

Reviewer #2: Yes

Reviewer #3: No

Reviewer #4: No

4. Is the manuscript presented in an intelligible fashion and written in standard English?

Reviewer #1: Yes

Reviewer #2: Yes

Reviewer #3: Yes

Reviewer #4: Yes

Reviewer #1: The authors are presenting a rodent study in which the want to describe the effect of Exercise Preconditioning on Myocardial Content of S1P and its Mechanism after Exhaustive Exercise. Unfortunately, the data presented is not comprehensive enough to seriously underline the authors' conclusions. There is a clear need for improvement both in the methodology and in the presentation and interpretation of the data. Furthermore, the knowledge gap to be filled here is very small. There is some data on S1P in exercise and training that should have been discussed in more detail here. Overall, the idea is a good one, but significant improvements are needed before publication. My suggestions are as follows:

Abstract:

- Typos in S1P

- Methods described in great detail in relation to the rest.

- The results are difficult to understand. The enumeration (1.,2.,3.) seems more like a bullet point list. Where possible, the wording should also be chosen so that results are compared with the control group. Otherwise this is very misleading.

Methods:

- The headings should be more clearly separated from the text. Reading would be easier if the headings were also visually marked (bold, italics, etc.).

- W146 is not just an “S1P receptor blocker”, but a specific antagonist of S1PR1. This should be made clear, as S1P can mediate its effects via the other S1PRs

- The description of PD98059 as a MAPK blocker is also not ideal. It is a MAPKK inhibitor (MEK1/2)

- It remains unclear in which solvent the substances were administered

- Were there any differences in the duration of the final “single bout of exhaustive swimming exercise” in the test animals?

- The preparation and the amount of myocardial tissue to be prepared for the S1P determination is not mentioned

Results:

- The authors refer to tables in several places. However, these cannot be found in the manuscript.

- The designation “group C” for the controls is confusing, especially as the figures show “Control”.

- Fig. 1A: A reduction in heart weights cannot be seen from the graph alone. Even with the caveat that this is not a significant change, the authors should not describe these heart weights of the groups as different.

- When describing and presenting data, one should always compare the treatment with the controls. This alternate presentation (e.g. Results to Fig. 1B) is misleading. Here it would be “were decreased in EP group...”

- Fig. 1B: Can the histological effect also be quantified?

- Fig. 1D: 75% apoptosis in the myocardium after a single swim seems almost implausibly high to me. Does this match the literature? Is there any way to explain this? The animals should hardly be viable after this training.

- Fig. 2A. It is unclear how this unit comes about. The amount of S1P in the myocardium cannot be given as ng/ml without further information and explanations.

- Fig. 2A: Are there also differences in plasma S1P?

- Fig. 2C: Is there a possibility of quantification? A change in cardiomyocyte rupture, as described by the authors, can hardly be detected.

- Fig. 2E: Was the inhibition of other S1PR also tested? S1PR3 in particular also seems to be of interest here

- Fig. 3A. A sectioned Western blot image is not sufficient to support the statement “the expression of p-MEK in the myocardium of rats... was significantly reduced...”. Quantification with statistically significant differences is required here.

- Fig. 3B/C: It is not possible to speak of differences here. The text should be revised.

Discussion

- The authors do offer a number of references to justify the finding mechanism. However, this is not sufficient. It is possible that the increased formation of S1P via SphK1 also plays a role here. This has already been shown for other cells that were also exposed to a certain amount of stress (Benkhoff et al., Circulation. 2025).

Reviewer #2: Strengths

The experimental design is logical and relevant to the study question.

The use of both S1P receptor and MAPK pathway inhibitors helps dissect the mechanisms involved.

The figures support the text well and show clear experimental differences.

Ethical approval and animal welfare guidelines were followed.

Weaknesses and Suggestions for Improvement

1. Sample Size & Randomization: Clarify how rats were randomized into groups. Indicate whether investigators were blinded during analysis.

2. Data Transparency: Include numerical data (mean ± SD) within the main text or supplementary tables. For instance, actual S1P concentrations in each group. Figures are provided, although raw data (e.g., actual ELISA values, full apoptotic counts) could further strengthen transparency.

3. Statistics: Indicate whether post hoc tests were used. Also, include effect sizes or confidence intervals where possible. The manuscript lacks a statement on statistical power or sample size calculation. Exact p-values (rather than simply “P < 0.05”) would improve transparency.

4. Discussion: While generally well-referenced, the discussion would benefit from addressing alternative pathways (e.g., eNOS or STAT3) as mentioned briefly at the end. Consider whether MAPK activation is a downstream result or a parallel pathway influenced by S1P.

5. Figure Legends: Improve clarity. Example: Figure 2 – instead of “showed increase,” write “was significantly higher than the EP group (P < 0.05).”

6. Grammar & Language: Minor editing needed for readability and fluency. A few examples:

Replace “cardiomyocyte rupture and interstitial changes were increased” with “cardiomyocyte rupture and interstitial edema were more prominent.”

“This study was carried out in accordance with the principles of the Basel Declaration.” → "The study adhered to the Basel Declaration on animal research ethics."

"An observable dysregulation of swimming actions..." → Consider simplifying to "evident swimming fatigue."

Use active voice more often. Avoid redundant phrases like “compared to the EP group, the EP+W146 group showed…” repetitively.

Figures and Tables

Figure Quality: Acceptable but can be improved by adding scale bars, clearer labels (e.g., “n = 6/group”), and including actual values in bar charts.

Table Clarity: The tables referenced (Table 1–8) ??? are not embedded or visualized in the version reviewed. If these exist separately, ensure they are submitted as part of the supplemental material.

Reviewer #3: Thank you to the authors for their comprehensive and well-designed study on exercise preconditioning and its effects on myocardial S1P content in rats. The study is timely, and it provides valuable insights into the potential mechanisms underlying exercise-induced myocardial protection. Below are my specific comments aimed at improving clarity, experimental rigor, and presentation of results.

Introduction

Introduction is written summarized and well. The only point that could be mentioned is the relationship of S1P and MAPK in the literature, regardless of preconditioning. Before writing " Previous reports indicate that the MAPK signaling pathway may attenuate cardiac fibrosis, apoptosis, oxidative stress injury, and inflammatory reactions [5]." You can bring a bridge from s1p to MAPK and their importance in preconditioning… .

Methodological Recommendations:

1. Clarification of Exercise Protocol:

o The manuscript states that the exercise duration gradually increased from 30 minutes/day to 2 hours/day. It would be helpful to specify how the intensity of the swimming exercise changed during this period. For example, was the swimming intensity (e.g., swim speed or resistance) progressively increased along with the duration? Detailing the intensity would improve the reproducibility of the exercise protocol.

2. Control Group and Randomization:

o While the animals were randomly assigned to groups, it is essential to explicitly mention whether the experimenters were blinded to the group assignments and the outcome measures (e.g., apoptosis index, S1P levels). Blinding helps reduce potential biases during outcome assessments.

3. Sample Size Justification:

o While the sample sizes for each group are mentioned (e.g., n=5 or n=6), the manuscript would benefit from a clear justification of how these sample sizes were determined. Were power analyses conducted to ensure that the study had enough statistical power to detect differences between groups? If not, this could be a limitation to address in the revised manuscript.

4. S1P Measurement:

o The manuscript mentions using an ELISA kit to measure myocardial S1P content. Did you validate the S1P ELISA kit against other methods, such as mass spectrometry, to ensure its accuracy? This validation would strengthen the reliability of the S1P data.

5. Inhibition Timing and Dosing:

o The manuscript describes the use of the S1P receptor blocker W146 and MAPK inhibitor PD98059, but the exact timing of administration relative to the exercise sessions could be more clearly defined. Were these inhibitors given before every exercise session or just once a day? Was the dose adjusted over time based on the animals' body weight? These details are critical for reproducibility and understanding the effectiveness of the treatments.

6. Method of Exhaustion Induction:

o The criterion for exhaustion (e.g., observable dysregulation of swimming actions, slow paddling speed, continuous sinking) is subjective. While this is commonly used in exercise studies, addressing this issue in limitation would enhance the clarity of the findings.

Results and Data Presentation:

1. Figures

o It would be informative to provide the exact data or statistical analysis in the figure legend to allow the reader to understand how significant the difference is between groups.

o In Figure 3E, apoptosis is increased in the EP + PD98059 group. A discussion about the mechanisms contributing to this finding (i.e., how MAPK inhibition could increase myocardial damage despite exercise preconditioning) would be helpful to provide a more comprehensive interpretation of the data.

Reviewer #4: Comment 1: The tables are not visible in the main text. Please include them at the end of the manuscript instead of submitting them as a supplementary file.

Comment 2: While the study suggests MAPK signaling is involved in S1P-mediated protection, the molecular mechanism remains speculative. Consider discussing alternative or complementary signaling pathways such as AKT or STAT3, which were briefly mentioned but not tested.

**Do you want your identity to be public for this peer review?** For information about this choice, including consent withdrawal, please see our Privacy Policy

Reviewer #1: No

Reviewer #2: No

Reviewer #3: No

Reviewer #4: No

---

## [Author Response · Author response to Decision Letter 1]

10 Sep 2025

We have made correction according to the Reviewer’s comments.

---

## [Decision Letter · Decision Letter 1]

29 Sep 2025

Dear Dr. Yang,

Thank you for submitting your manuscript to PLOS ONE. After careful consideration, we feel that it has merit but does not fully meet PLOS ONE’s publication criteria as it currently stands. Therefore, we invite you to submit a revised version of the manuscript that addresses the points raised during the review process.

We look forward to receiving your revised manuscript.

Kind regards,

Amirreza Khalaji

Academic Editor

PLOS ONE

Journal Requirements:

Reviewers' comments:

Reviewer's Responses to Questions

**Comments to the Author**

Reviewer #3: All comments have been addressed

Reviewer #5: (No Response)

Reviewer #6: All comments have been addressed

Reviewer #7: All comments have been addressed

Reviewer #8: (No Response)

2. Is the manuscript technically sound, and do the data support the conclusions?

Reviewer #3: Yes

Reviewer #5: No

Reviewer #6: Yes

Reviewer #7: Yes

Reviewer #8: Yes

3. Has the statistical analysis been performed appropriately and rigorously?

Reviewer #3: Yes

Reviewer #5: No

Reviewer #6: Yes

Reviewer #7: Yes

Reviewer #8: Yes

4. Have the authors made all data underlying the findings in their manuscript fully available?

Reviewer #3: Yes

Reviewer #5: No

Reviewer #6: Yes

Reviewer #7: Yes

Reviewer #8: No

5. Is the manuscript presented in an intelligible fashion and written in standard English?

Reviewer #3: Yes

Reviewer #5: No

Reviewer #6: Yes

Reviewer #7: Yes

Reviewer #8: Yes

Reviewer #3: Dear authors,

Thank you for applying the recommendations. The current version of the manuscript is clearer and more rigorous.

Best wishes

Reviewer #5: The manuscript investigates whether exercise preconditioning (EP) mitigates exhaustive-exercise–induced myocardial injury in rats and explores a mechanistic pathway involving sphingosine-1-phosphate (S1P) and MAPK signaling across four groups (Control, EP, EP+W146, EP+PD98059), with outcomes including heart weight, histology, apoptosis, S1P levels, and p-MEK. This is a revised submission; however, substantive issues remain in methodological reporting (e.g., blinding, euthanasia parameters, randomization, sample-size justification), statistical transparency (tests, assumptions, multiple-comparison control, effect sizes/CIs), and internal consistency between text, tables, and figure legends. Several contradictions (e.g., non-significant results described as decreases) and formatting/artifact problems persist, affecting interpretability and reproducibility. In its current form, the study requires major revision to meet PLOS ONE’s rigor and reporting standards.

Major comments

1. Results describe a decrease “not statistically significant,” and the table shows no difference (e.g., p≈0.95), yet the Figure 3 legend asserts a decrease. All sections must agree and use neutral wording that reflects the statistics.

2. Results indicate no significant difference (P>0.05), but the figure legend states “Cardiac weight was increased.” Legend phrasing should mirror the statistical outcome.

3. Table content contains a merged/duplicated line and mixes mean±SD with median(IQR) for the same endpoint. Clean the table and use one summary metric aligned with the applied test.

4. CO₂ flow is reported as “3–70% chamber volume per minute,” and elsewhere euthanasia is described under deep anesthesia “when necessary.” Specify one guideline-concordant method with exact parameters and apply it consistently.

5. Blinding is stated despite visibly different procedures. Specify who was blinded, how (coding/masking), and which outcomes were assessed blinded (e.g., coded histology/apoptosis); otherwise revise to assessor-blind.

6. Power is reported using a regression effect size (f²=0.15) for a four-group design; achieved n≈5–6/group is small for histology/apoptosis. Provide an ANOVA/GLM-appropriate calculation (primary outcome, effect size, α, power) or state that the study is exploratory.

7. P-values are reported without naming the tests, assumption checks, multiple-comparison handling, effect sizes, or 95% CIs. Add a Statistics subsection with software, tests/models per endpoint, diagnostics, and corrections; report estimates with CIs.

8. Clarify whether heart weight is absolute or normalized (and how), define apoptosis index quantification (fields, thresholds, blinding), provide S1P kit/catalog details, and include n/group in figure panels.

9. Randomization is stated without method/timing (sequence generation, concealment, independent allocator). Provide concrete implementation steps.

Minor comments

10. Figure legends—units and n/group: Add units (e.g., S1P ng/mL, °C) and n/group in each legend; ensure legend wording reflects the actual statistical outcome (e.g., “no significant difference” where applicable).

11. Data availability / ethics alignment. Ensure the Data Availability and Ethics statements match Methods with precise institutional approval details and access to raw per-animal data (and original images where applicable).

Reviewer #6: (No Response)

Reviewer #7: Thank you for the opportunity to review; the manuscript has been carefully revised—no further changes are required.

Reviewer #8: Dear Authors

Thank you for your submission and for the substantial effort invested in a valuable. Yor revision sounds good and almost complete, but there are minor things to care. The comments that follow are offered in a constructive spirit to help sharpen the manuscript’s clarity and maximize its usefulness to researchers.

My comments:

-Reconcile the direction of S1P change (EP vs Control) across text, tables, and figures; ensure the narrative matches the numeric values and final conclusions.

-Consolidate to one Statistics section (remove legacy one-way ANOVA text if not used).

-Name a single primary endpoint (used for power) and map each outcome (heart weight, S1P, apoptosis, p-MEK) to: -distribution check (e.g., Shapiro–Wilk),-summary measure (mean±SD or median[IQR]), exact test used, post-hoc method for the 4-group design, multiple-comparison control (e.g., Tukey, Dunn, Holm/Benjamini–Hochberg), effect sizes and 95% CIs.

-Resolve software(version) inconsistencies (e.g., SPSS 17 vs newer; Prism) and state the final versions only.

-Use W146 = S1PR1-selective antagonist; PD98059 = MEK1/2 (MAPKK) inhibitor.

Remove any remaining “MAPK blocker” wording in text, tables, and figure legends.

-Report myocardial S1P normalized to tissue mass (ng/g) or protein (ng/mg); if retaining ng/mL homogenate, provide a clear calculation pipeline (homogenization volumes, protein assay) so others can reproduce it.

Add operational details, who generated or kept the randomization list, how allocation was concealed, who assigned animals, who was blinded for TUNEL scoring and Western densitometry, and when blinding was lifted.

Reconcile inconsistent euthanasia, and anesthesia descriptions; ensure the Methods reflect the actual procedure and are consistent across manuscript and submission materials.

-Include representative images with scale bars.

-Describe field selection, counting thresholds, automation, and positive-negative controls to address potential TUNEL over-calling.

-Remove tracked-changes artifacts, duplications; correct any duplicated or corrupted table entries.

-Ensure each legend lists n/group, exact test, post-hoc method, and exact P-value criteria.

- Add explicit statements on: subjective exhaustion criterion; small sample size; single sex; ELISA vs LC-MS/MS for S1P; lack of functional cardiac readouts (e.g., echo, infarct size); and inhibitor specificity caveats.

-Specify exact timing of W146 and PD98059 (relative to each preconditioning session and to the exhaustive swim), vehicle, dose per kg, and whether dosing changed over time.

-State clearly what is novel (S1P→MAPK inhibition within an exercise-preconditioning model) versus confirmatory, and situate findings against the most relevant prior EP/S1P cardioprotection literature.

**Do you want your identity to be public for this peer review?** For information about this choice, including consent withdrawal, please see our Privacy Policy

Reviewer #3: No

Reviewer #5: No

Reviewer #6: No

Reviewer #7: No

Reviewer #8: No

---

## [Author Response · Author response to Decision Letter 2]

29 Oct 2025

We are deeply grateful to Dr. Amirreza Khalaji and the Reviewers for their time and insightful comments on our manuscript. We have carefully considered all feedback and have thoroughly revised the manuscript to address each point raised.

Key improvements include:

Explicitly framing the study as exploratory and clarifying the novel S1P→MAPK signaling hypothesis versus confirmatory aspects in the Discussion.

Adding a detailed “Statistical Analysis” subsection and ensuring comprehensive reporting of tests, effect sizes, and confidence intervals throughout.

Enhancing methodological transparency regarding heart weight, apoptosis quantification, S1P normalization, and sample size indication in figures.

Consulting a biostatistician to ensure the rigor and appropriateness of all statistical approaches.

We believe these revisions have significantly strengthened the manuscript and hope it now meets the journal’s standards for publication. Thank you again for the opportunity to improve our work.

---

## [Decision Letter · Decision Letter 2]

11 Nov 2025

Dear Dr. Weiyuan Yang,

Thank you for submitting your manuscript to PLOS ONE. After careful consideration, we feel that it has merit but does not fully meet PLOS ONE’s publication criteria as it currently stands. Therefore, we invite you to submit a revised version of the manuscript that addresses the points raised during the review process.

We look forward to receiving your revised manuscript.

Kind regards,

Amirreza Khalaji

Academic Editor

PLOS ONE

Journal Requirements:

Reviewers' comments:

Reviewer's Responses to Questions

**Comments to the Author**

Reviewer #5: (No Response)

Reviewer #8: (No Response)

2. Is the manuscript technically sound, and do the data support the conclusions?

Reviewer #5: No

Reviewer #8: Partly

3. Has the statistical analysis been performed appropriately and rigorously?

Reviewer #5: No

Reviewer #8: N/A

4. Have the authors made all data underlying the findings in their manuscript fully available?

Reviewer #5: No

Reviewer #8: Yes

5. Is the manuscript presented in an intelligible fashion and written in standard English?

Reviewer #5: No

Reviewer #8: Yes

Reviewer #5: The revision does not provide proper, point-by-point answers to the reviewers’ comments; many substantive issues remain unaddressed. For example, a required Declarations section is still missing (ethics approval/consent, data availability with an accessible link, funding, competing interests, and author contributions).

Reviewer #8: Dear authors,

I checked all revision files, unfortunately I can't find point by point response letter or section from authors.

please upload it again.

**Do you want your identity to be public for this peer review?** For information about this choice, including consent withdrawal, please see our Privacy Policy

Reviewer #5: No

Reviewer #8: No

---

## [Author Response · Author response to Decision Letter 3]

12 Nov 2025

Reviewer #5: Thank you for your feedback and for bringing this to our attention. We sincerely apologize for the oversight in our previous submission. The point-by-point response to the reviewers' comments was inadvertently omitted during the upload process. We have now uploaded the "Response to Reviewers" document to the submission portal. This document provides a detailed, point-by-point account of all changes made in response to the reviewers' valuable comments. Regarding the Declarations section, we confirm that all required statements, including the ethics approval and consent which are detailed in section 1.2.4 "Ethics Statement" of the manuscript, are fully included. The complete Data Availability statement with the accessible link is also present within the main manuscript file.

We regret any confusion or inconvenience this omission may have caused. Thank you for your patience and the opportunity to correct this error.

Reviewer #8: Please accept our sincere apologies for this oversight. The point-by-point response letter was inadvertently omitted. We have now uploaded the file "Response to Reviewers_updated.doc" to the submission portal. Thank you for your patience and for allowing us to correct this error.

---

## [Editor Report · Decision Letter 3]

18 Dec 2025

Effect of Exercise Preconditioning on Myocardial Content of Sphingosine1-phosphate and its Mechanism in Rats after Exhaustive Exercise

PONE-D-25-14601R3

Dear Dr. Weiyuan Yang,

We’re pleased to inform you that your manuscript has been judged scientifically suitable for publication and will be formally accepted for publication once it meets all outstanding technical requirements.

Kind regards,

Amirreza Khalaji

Academic Editor

PLOS One
---

## [Editor Report · Acceptance letter]

PONE-D-25-14601R3

PLOS One

Dear Dr. Yang,

I'm pleased to inform you that your manuscript has been deemed suitable for publication in PLOS One. Congratulations! Your manuscript is now being handed over to our production team.

Kind regards,

on behalf of

Dr. Amirreza Khalaji

Academic Editor

PLOS One